# Communicating Science through Comics: A Method

**Jan Friesen [1,2,*] 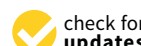, John T. Van Stan II [3] and Skander Elleuche [1,4,*]**

1    Arab-German Young Academy of Sciences and Humanities (AGYA), Working group "Energy, Water and Environment", at the Berlin-Brandenburg Academy of Sciences and Humanities (BBAW), Jägerstr. 22-23, 10117 Berlin, Germany
2    Department of Catchment Hydrology, Helmholtz Centre for Environmental Research—UFZ, Permoserstrasse 15, 04318 Leipzig, Germany
3    Geology and Geography, Georgia Southern University, 68 Georgia Ave, Statesboro, GA 31401, USA; jvanstan@georgiasouthern.edu
4    Miltenyi Biotec GmbH, Friedrich-Ebert-Straße 68, 51429 Bergisch Gladbach, Germany
*    Correspondence: jan.friesen@ufz.de (J.F.); skander.elleuche@rub.de (S.E.)

**Abstract:** Scientists are trained to tell stories, scientific stories. Training is also needed to comprehend and contextualize these highly nuanced and technical stories because they are designed to explicitly convey scientific results, delineate their limitations, and describe a reproducible "plot" so that any thorough reenactment can achieve a similar conclusion. Although a carefully constructed scientific story may be crystal clear to other scientists in the same discipline, they are often inaccessible to broader audiences. This is problematic as scientists are increasingly expected to communicate their work to broader audiences that range from specialists in other disciplines to the general public. In fact, science communication is of increasing importance to acquire funding and generate effective outreach, as well as introduce, and sometimes even justify, research to society. This paper suggests a simple and flexible framework to translate a complex scientific publication into a broadly-accessible comic format. Examples are given for embedding scientific details into an easy-to-understand storyline. A background story is developed and panels are generated that convey scientific information via plain language coupled with recurring comic elements to maximize comprehension and memorability. This methodology is an attempt to alleviate the inherent limitations of interdisciplinary and public comprehension that result from standard scientific publication and dissemination practices. We also hope that this methodology will help colleagues enter into the field of science comics.

**Keywords:** academic publishing; comics; popular science; science communication; sequential art

## 1. Introduction

The language of scientists and their stories, even at its best, can often be compared to Tolkien's Old Entish, described by an Entish native speaker as "a lovely language, but it takes a very long time saying anything in it, because we do not say anything in it, unless it is worth taking a long time to say, and to listen to." This is because scientists must describe every meticulous detail of their controlled and repeatable experiments, how these experiments target specific processes to evaluate nuanced hypotheses and, hopefully, report new, exciting, often incremental findings that are then published in scientific journals after rigorous peer review [1]. Nowadays, it is becoming increasingly important for scientific endeavors to attract awareness and to reach a broad audience [2]. In the scientific community, ways to simplify science are implemented by, for example, adding graphical abstracts or AudioSlides to papers that are aimed at providing a quick, concise overview of the presented work [3]. Moreover, the Journal of Visual Experiments (JoVE) even publishes videos of experiments in combination with short method papers to visualize laboratory or field experimental work.

However, the science itself is often rather complex, which necessitates that it be generalized and simplified. Common, widely-used approaches to communicate science are TED talks [4], Science Slams [5], educational YouTube channels, such as MinuteEarth or plantBRUTALITY [6,7], and sequential art, or comics [8–11]. In the past years, comics and graphic novels have experienced a substantial rise as based on growth rates from the publishing sector [12]. The choice of a scientific communication format depends on the target audience, often an audience that is not covered by those who typically consume scientific publications. Nevertheless, it is very important that the consumer is not just gaining knowledge of the result, but of the underlying methods that produced the result and its uncertainties. The presented approach, as such, can be aimed at different target groups or members of the general public [13] as we do not specify what level of detail or how much technical terms should be included. In general, however, the aim with comics is to transport a substantial amount of information through imagery and, in our case, we provide an opportunity for the audience to delve deeper into the topic by providing the reference to the scientific article. Thus, it can be argued that our audience starts at high school level depending on the motivation of the reader to work themselves through a scientific article or look up specific technical terms.

Scientific papers are often motivated by global relevance such as food security, health, or energy demands. Popular examples of illustrated scientific works are, for example, the series "Science comics—get to know your universe" [8], the extensive graphic novels on the genesis of our planet, the evolution of life and the history of humans by Jens Harder [9,10], or the series "Serious Scientific Answers to Absurd Hypothetical Questions" by Randall Munroe [11]. Even renowned scientific journals, such as Nature and organizations such as the US National Aeronautics and Space Association (NASA), make use of scientific illustrations, for example, to summarize 25 years of climate problem discussions [14] and to explain remote sensing satellites for precipitation monitoring [15]. Another example from the scientific community is a book covering different topics in biotechnology and medicine, which is highlighted by the illustration of a tiny alien, capable of travelling in the human bloodstream or even in cells explaining principles of biocatalysis and diabetes [16]. Moreover, one of the most iconic artists in the world of graphic novels and comics, Will Eisner, educated soldiers through comics during World War II and went on producing technical bulletins in comic-style until 1971 [17]. Just this year, Farinella [18] mentions the "emerging field of 'graphic science'" and discussed the potential of comics in science communication.

An extensive overview of science comics has been provided by Tatalovic [19] who not only defines the field, but also provides a review of several science comics. Science comics are commonly used for education, as well as for science communication that focuses on outreach and awareness raising [13,19]. In terms of science comics aimed at education, Aisyah et al. [20] and Tribull [21] provide guidelines. Aisyah et al. [20] describe the development of a comic on crude oil for school education including educational elements such as questions that are presented in a gamified structure. Tribull [21] further includes guidelines regarding distribution and funding, as well as impact assessment. Besides methodological studies on how to make science comics, in the field of education, there are also several studies that show their educational impact and benefit by incorporating comics in teaching curricula [22–24]. The motivation for the presented approach is primarily in outreach and awareness raising [19], not in education. However, the presented methodology is easily extendable to craft educational content or elements (such as questions) and, in terms of language, it can be extended to suit specific target groups or age classes.

Comics are utilized to cover important and complex issues in an appealing and interesting format that might introduce a new audience to scientific topics. Scientific background can be textbook knowledge, based on multiple sources or single publications. In this paper, therefore, we developed a methodology specifically aimed at natural scientists to provide step-by-step instructions on how to conceptualize and design a comic based on an individual scientific publication. This includes the development of a concept, a detailed storyline, and versatile characters. This method has been initially developed and applied to make a comic based on an interdisciplinary review paper on halophytes and

bioenergy [25,26] (Figure 1). Our methodology is flexible and the presented framework can easily be adapted to convert scientific publications from different disciplines into a comic format.

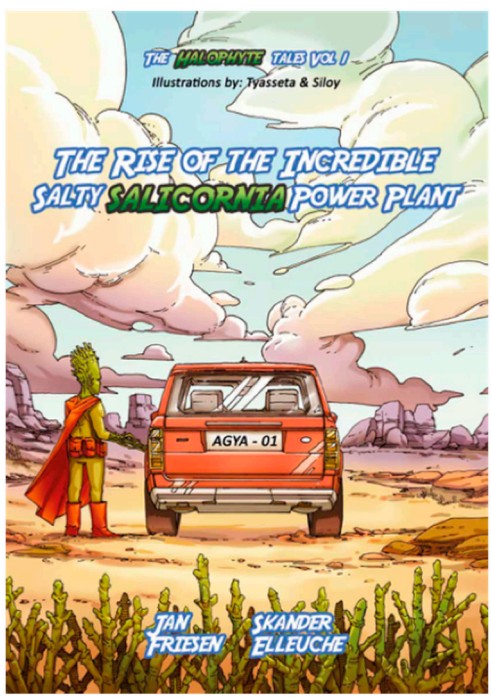

**Figure 1.** The cover of the first episode of the "Halophyte tales" comic series.

## 2. Four Steps to Produce a Science Comic

In the natural sciences, most findings nowadays are communicated through scientific publications, often in the form of relatively concise papers of 10–20 pages. In our approach, we use single publications instead of multiple sources and convert them into science comics. The main objective of science comics is to transport the full scientific methods, findings, and uncertainties into an easily understandable format. Another aim is to provide ample scientific background to enable readers to delve further into the topic. Transforming scientific articles into comics, in general, requires one to (i) communicate most content through sequential images, and (ii) reduce the scientific details to very short and concise messages that clearly bind the sequential imagery together into a linear narrative. Successfully executing these transformative processes requires scientists to reduce expert knowledge to a level that allows the inexperienced audience to understand complex topics in an attractive way.

To produce such a comic, we present a process that is split into four "developmental" subdivisions: (i) develop a conceptual foundation; (ii) develop a setting that graphically ties important scientific elements to the conceptual foundation; (iii) develop characters that graphically describe the science; and (iv) develop a detailed storyline that weaves together the conceptual foundation, setting, and characters into a linear narrative.

### 2.1. How to Develop a Conceptual Foundation?

The term conceptual foundation, in this case, is a mental representation of the scientific endeavor contained within any individual technical publication. The process of developing a conceptual foundation for a science comic begins with extracting the critical elements that shaped (and limited) the scientific endeavor and, along the way, separating these elements from subsidiary information. A good method for this is to summarize the research in highlights or bullet points and focus on as few as possible (i.e., like "research highlights" currently required by many natural science journals). The conceptual foundation should then seek to unify these research highlights. Once a concept is

established, scientists can use this broad mental representation to identify a range of related recurrent or significant themes that are broadly relatable across the human experience. Recurrent and significant themes related to the foundational concept often inherently include fictional components (storylines, characters, and settings) that can be valuable vehicles for conveying facts. For example, research highlights [25] to include, are:

- Population and economic pressures require development of renewable energy sources;
- Plant-derived bioenergy sources are promising, but compete with conventional crops;
- Salt-tolerant plants, halophytes, do not compete with conventional crops;
- Halophilic (salt-loving) microbes may speed up degradation of halophyte plants into bioenergy;
- Feasibility of halophyte-derived biofuel as a mid-term renewable energy source is assessed.

An example conceptual foundation for a comic based on these research highlights emerges: Can halophytes fuel the world? This builds off mental representations that are broadly relatable across the human experience and, naturally, tap into fictional tropes: like superheroes (i.e., "Can Superman save the world?"), apocalyptic events that threaten dystopian futures (i.e., "What happens if the world runs out of fuel?"), and the "everyman" (or everywoman) and underdog concept (i.e., "Even the oft-ignored and isolated halophyte can save the world!"). The superhero trope inspired panels representing the halophyte superhero ("The halophyteman") in Figure 2.

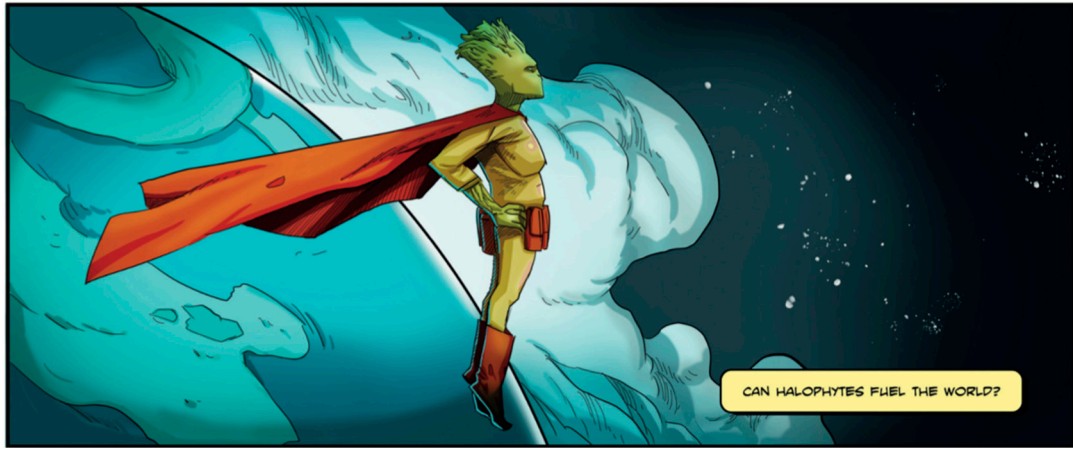

**Figure 2.** The "superhero" trope inherent in the concept "Can halophytes fuel the world?".

## 2.2. Development of a Scientifically-Relevant Setting

The setting must be rooted in the conceptual foundation and must be "scientifically relevant," in that it is used to motivate the science. Natural sciences research is often necessitated by real-world problems, such as energy demand, for which a solution is sought. This motivation can be used to identify the settings that surround common stakeholders who are affected by the topic. In our example, the conceptual foundation derived from our review paper is the potential of halophytes to be used as an alternative energy resource [25]. Since halophytes do not compete with food crops, because they grow on salty soil, these plants may be a valuable resource for the production of bioethanol. Common physiographic settings that contain stakeholders interested in this would, therefore, typically include semi-arid to arid regions with a supply of salty soils, for example, desert environments. Human geographic components can be overlaid atop these physiographic features to help further hone the setting within which stakeholder interest is high. Looking at the various desert areas of the world, are there any that contain particular socioeconomic conditions that likely enhance stakeholder interest in renewable energy, like an undiversified economy that is heavily reliant on non-renewable energy? Arguably, the most scientifically-relevant setting, rooted in our example conceptual foundation, clearly exists at the intersection of desert conditions and fossil fuel-driven economic conditions.

As a result of this, we adapted the example review article into a research comic set in an oil-based desert state whose ruler (or, stakeholder) has clear interests in investment in renewable energy. To realize this, he invites four young scholars from interdisciplinary research fields to develop a plan for the production of green energy in a fictitious Arab megacity. Moreover, the setting can assist in the avoidance of technical terms via attractive ways to explain complex contexts (i.e., by picturing laboratory settings or scenes of natural phenomena).

### 2.3. Development of Characters

For the actual research that is embedded in the conceptual foundation and setting of the story, characters need to be designed that depict the major scientific elements and endeavors. Based on the conceptual foundation, one may re-adjust the focus of the original research publication to develop an easy-to-follow, exciting, and humorous comic story. Humor especially sets free emotional energy, often leading to positive feedbacks. Such feedbacks can both enhance reading motivation as well as speed [27]. Therefore, all information must be covered graphically or in the form of speech bubbles.

The characters play an important role in the emotionalization of science communication [28]. The audience can be influenced, because a comic illustration of characters linked to broadly relatable concepts might wake emotions in the reader. Therefore, it is important to design characters that enhance the scientific story, depict certain details related to scientific context, methods, etc., or have specific functions that further or deepen the narrative. Iconic characters can help embed a complex scientific problem or process in a storyline [29]. Additionally, technical terms that are essential for the story must therefore be de-mystified to keep the drive of the narrative style. This de-mystification can be done en route by various characters, or it may be more efficient (and improve the story's fluidity) if an individual character is developed that de-mystifies all the unavoidable technical terms—a science communicator.

In our example [26], a sultan and a "science jinn" (Figure 3) were developed to build a pair of friendly but contradicting buddies. The sultan is a well-tempered and a wise ruler of a fictitious, Arabic realm. He has a great interest in sustainable technologies to economically maintain his megacity and the beautiful environment in the future. He is giving important background information in colloquial language. However, the science jinn is willing to explain disciplinary terminology and to bridge disciplinary boundaries to finally attain solutions. Ultimately, the science jinn is a science communicator, who translates expert knowledge from different disciplines in an interdisciplinary research team. He is treated as being omniscient, but, in our example, to provide character, he also is portrayed as a real fault-finder and even interrupts the sultan, because he always knows best. It is challenging to develop a likeable character that is all-knowing, or at least enough-knowing, to explain every technical term. As a result, traits for the humanization of the science communicator character are recommended, but will naturally vary depending on the relationship that character has with others.

An example of the science jinn's capability to overcome interdisciplinary communication barriers includes: a character drinks sweet tea in one scene and says: "The glucose in the tea is directly imported into the mitochondria in my body cells!" The science jinn translates for the non-expert reader that "Sugar is pure energy. Plant cell walls are mainly composed of sugars". This is a strategy to use characters and settings to visually convey scientific concepts or experiments in sequential art. We illustrated the energy boost effect by means of a power flash and a mirrored image of the scientist (Figure 4) that is a reminiscence to Asterix' magic potion effect. Other examples to illustrate processes may be Popeye's spinach for energy boosts or Speedy Gonzales for speedy processes. In addition, some dummy or side story characters can be employed to add practical jokes. In our comic, the sultan always mentions his mother in law, but she never shows up.

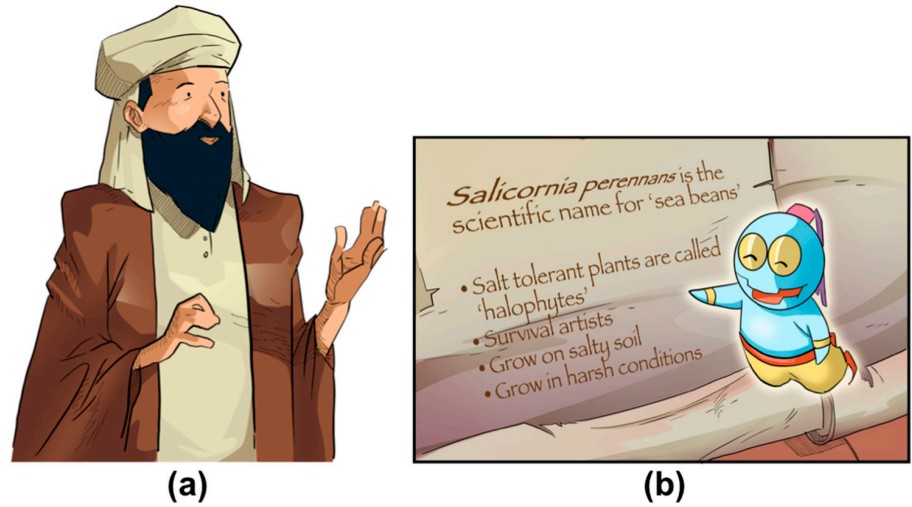

**Figure 3.** The sultan (**a**) and the science jinn explaining the technical term "halophytes" (**b**).

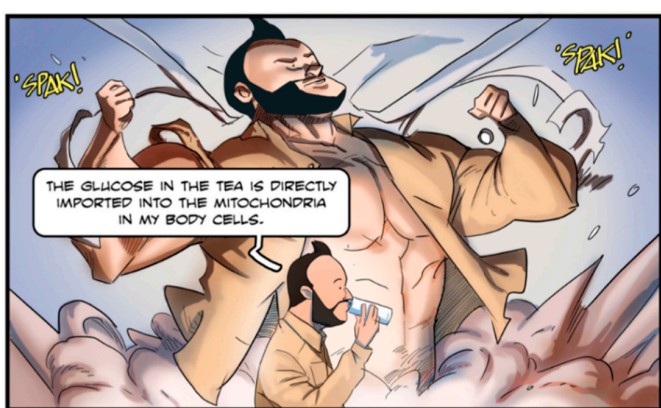

**Figure 4.** Power flash moment.

In a science-based setting, another option would be to use the study authors as characters and their different involvement or discipline. A depiction of different disciplines may be realized by specific work clothes, such as lab coats, outdoor clothes for field work, or boiler suits for engineering sciences. Such a stereotypic representation of occupation or of specific processes in an iconic manner is used to reduce text and is therefore also referred to as "part of the language of graphic storytelling" [29].

In our case, the scientific contribution and content of each author-based character has been reduced to one page per discipline. Each discipline is highlighted on a "lab" page that depicts the work done within the four different disciplines, without giving further scientific detail. The disciplines themselves are depicted through iconic equipment, such as petri dishes, microscopes, or greenhouses.

Despite the depiction of different characters, it is important to use as much imagery as possible instead of using long explanatory texts in the form of speech bubbles or text boxes. As an example, this was realized in our comic by using the imagery of large, fuel-guzzling SUVs and through plants being pushed into the fuel tank as a pictogram for bioenergy (Figure 5).

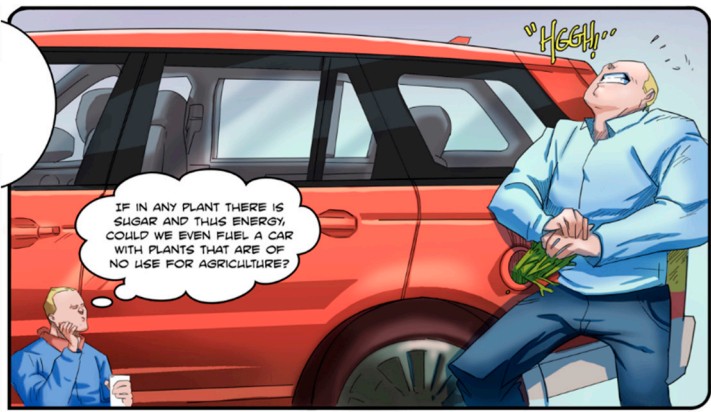

**Figure 5.** Fuel-guzzling SUV being fueled with halophytes.

## 2.4. Development of a Detailed Storyboard

In our approach, we work with artists that realize our concepts. The artists are mostly scientific laymen, neither familiar with the specific scientific discipline nor scientifically trained, which makes a detailed storyboard very useful. Moreover, a detailed storyboard also helps when drawing the comic yourself. To produce a first draft, a good strategy might be to work with sample images (e.g., Google image search) of scientific equipment or specific settings to be used in the final drawing.

Such a storyboard should include the theme of each page, a description of the different panels including the envisioned imagery, and the text parts, which may be small text boxes, but are mostly speech bubbles or headers (Figure 6). The detailed storyboard itself is also a way to structure each page in terms of its main theme. Usually each page, although consisting of several panels, has a main theme and results in a message and/or a transition to the next page, similar to the end-focus technique in writing [30].

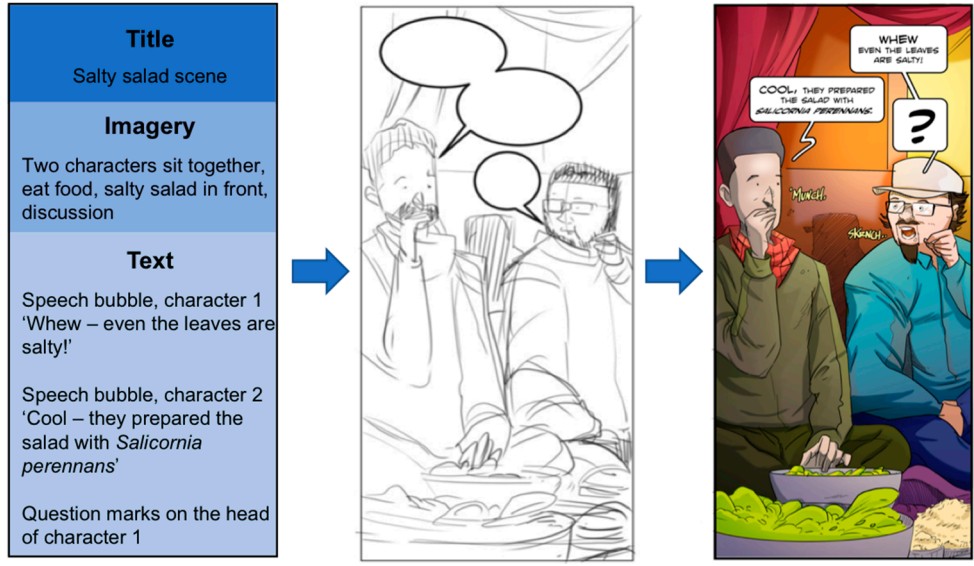

**Figure 6.** Exemplary page evolvement from detailed storyboard to sketch to final artwork.

In our example, the storyboarding began with the four young scholars hearing and discussing the Sultan's energy and economic concerns. They then come up with the idea that salt-loving plants, the so-called halophytes, might be valuable to be a resource for biofuel. The young scientists travel back to their labs and try to solve their task from four different points of view based on their respective disciplines. The reader gets some insights on the four different disciplines, without getting any detail,

but the comic strongly refers to the scientific research paper. At the end they present a concept to the sultan, who might enter a more sustainable era in his realm. In this way, and through this example, the background story is meant to envelope the science by introducing the topic to the reader and to provide a take-home message at the end of the comic.

Following the detailed storyline, it is important to develop a catchy title. The title of our comic "The rise of the incredible salty Salicornia power plant" is ambiguous, as *Salicornia* is the genus name of a plant and at the same time *Salicornia* is a potential bioenergy source to be used in industrial power plants. Moreover, the wording "the rise of . . . " is typical for superhero comics, action movies or computer games, such as "X-Men—The Rise of the Apocalypse", "Rise of the Planet of the Apes" or "The Rise of Tomb Raider". The adjective is also typical to describe superheroes like "The Incredible Hulk", or "The Incredibles".

As a way to increase the comic-like impression and to establish a brand, comic series titles and recurring characters can be used. In our example we introduce the series "Halophyte tales," which is a reminiscent of a well-known comic series, such as "Tales of the Unexpected," "Tales of Suspense," or "Tales from the Crypt". Characters, such as the science jinn, even have the potential to be used in further science comics irrespective of the topic as they have a general function to explain complex technical details, allowing scientific nomenclature to appear beside layman terms (a useful "decoder" for any layman interested in digging deeper into the scientific topic).

## 3. Discussion

The presented method is aimed at scientists that want to start making comics based on their research results. We do not aim to compete with professional artists or science communication experts. It is intended to serve as a step-by-step instruction for scientists that want to communicate their output in a different format. This subjective methodology is based on our personal experiences and has been developed during the production of our science comic [26]. In the meantime, this methodology has been successfully applied to produce a new comic "Urban forestry—Taming Precipita," [31] based on a recent book chapter [32] (Figure 7). Thereby, we develop the comic according to a specific scientific publication and reframe the scientific content in a way that it can be understood by everyone. The comic platform will raise awareness for such a topic, while the clear connection with a scientific publication further gives readers the opportunity to inform themselves on the "hard" science.

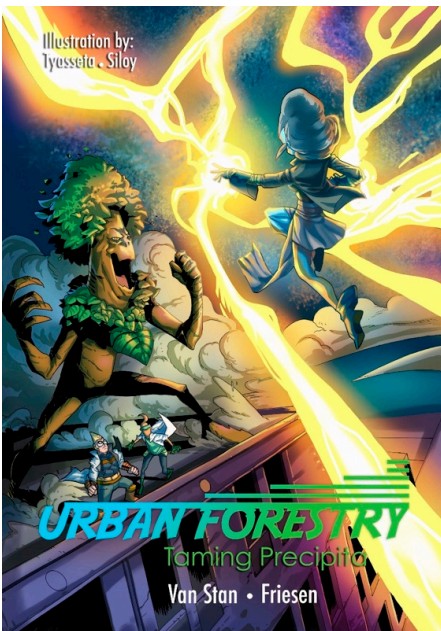

**Figure 7.** The cover of the forthcoming comic "Urban forestry: Taming Precipita".

Presenting science in the form of a comic is, besides our personal enthusiasm for comics, graphic novels, and art books, a way of reaching a larger, growing, and new audience for scientific topics. Nowadays, it has become possible to increase visibility [33] using social media, such as YouTube, Reddit (r/Science), Facebook, science blogs, Twitter, ResearchGate, or LinkedIn. Therefore, scientific communication is often given as open educational resources (OERs), e.g., in scientific cartoons, stop motion videos, online lectures, or Massive Open Online Courses (MOOCs). A "light" refurbishment of a complex topic might even be used to reach a younger audience. Cartoon illustrations in combination with a linear and simple narrative structure have been successfully demonstrated by the journal "Frontiers for Young Minds" to be read by young students at an age of 8–15. These articles are not only guaranteed to be written for kids, they are even edited by kids [34].

## 4. Conclusions

We have presented a clear and concise step-by-step methodology on how to convert single scientific publications into a science comic. The method is explained using an already published review article/comic pairing. Although developed subjectively, based on our own experiences, the method is flexible and provides a simple framework that can be adapted to individual preferences. We provide a pragmatic methodology that will hopefully motivate further colleagues that are eager to produce their own science-based comics.

**Author Contributions:** J.F. and S.E. conceived and designed the concept and wrote the paper. J.T.V.S.II reviewed and edited the paper.

**Funding:** J.F. and S.E. have received support from the Arab-German Young Academy of Sciences and Humanities (AGYA) that has been funded under the German Ministry of Education and Research (BMBF) grant 01DL16002. J.T.V.S.II received funding from the National Science Foundation (US-NSF) grant EAR-1518726.

**Acknowledgments:** This publication and the preparation of cartoon illustrations by comic artist and graphic designer Albertus Tyasseta (https://tyasseta.artstation.com) was supported by the Arab-German Young Academy of Sciences and Humanities (AGYA) and the Federal Ministry of Education and Research (BMBF).

**Conflicts of Interest:** The authors declare no conflict of interest. The funding sponsors had no role in the design of the study; in the collection, analyses, or interpretation of data; in the writing of the manuscript, or in the decision to publish the results.

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
