# Peer review of "Communicating Science through Comics: A Method"

_publications, doi:10.3390/publications6030038_

Round 1

Reviewer 1 Report

The authors describe a framework to translate scientific papers into a comic format, in order to disseminate scientific knowledge to a broader audience. They present a conceptual foundation using an example comic that is based on their own work.

While the scope of the work and the concept presented is from high interest for scientists who strive to communicate their work to the public, it lacks theoretical foundation. The presented framework is visualized by an example, but the several steps not constituted by evidence or literature. Even when considering the original and novel character of the framework, appropriate substantiation for the single steps (like in line 153-154) is needed. Additional literature from the field of science communication could be helpful, for instance

Spiegel, A. N., McQuillan, J., Halpin, P., Matuk, C., & Diamond, J. (2013). Engaging teenagers with science through comics. Research in science education, 43(6), 2309-2326

or

Tatalovic, M. (2009). Science comics as tools for science education and communication: a brief, exploratory study. Jcom, 8(4), A02.

These are just examples. There is way more literature from the field that could (and should) be considered.

Additional literature can help to outline concepts that are mentioned in the text, but not defined or proved, like the uncertainty or tentativeness of scientific results, or the usage of humor in the comic.

It could also be helpful to define the target group of the science comic, so appropriate tools and methods to reach that target group can be derived.

The framework itself should be described in a more abstract way, outlining general concepts. The sample comic is useful to illustrate aspects of the framework, but a scientific publication should go beyond of the mere description of a single case example.

Aim and idea of the work are interesting, but I strongly recommend working out a more substantiated scientific background.

Author Response

See attached Word file.

Reviewer 2 Report

Appreciate your enthusiasm for the form and willingness to dive in. I think the piece encourages the curious but nervous novice to get started trying comics. A more comprehensive piece would acknowledge different approaches besides the primary one discussed (even while still focusing on that particular method) but i'm not sure this needs to be comprehensive. As long as it's clear that this is your method - then that seems sufficient. 

Author Response

See attached Word file

Round 2

Reviewer 1 Report

In comparison to the initial submission, the authors substantially improved their work based on the reviews. I totally understand and appreciate the initial motivation of their work, which is not to write a fully comprehensive paper on science comic research, but to present a single case example as motivation and inspiration.

However, as this is a scientific journal, a certain level of profoundness must be fulfilled. Although the paper should meet that demand with the latest improvements, there would still be potential for a better conncetion to prior research and existing work on this field.